# miRNA-21 regulates CD69 and IL-10 expression in canine leishmaniasis

**Jaqueline Poleto Bragato, Gabriela Torres Rebech, Jéssica Henrique de Freitas, Marilene Oliveira dos Santos, Sidnei Ferro Costa, Flavia de Rezende Eugênio, Paulo Sérgio Patto dos Santos, Valéria Marçal Felix de Lima***

Department of Animal Clinic, Surgery and Reproduction, São Paulo State University, School of Veterinary Medicine, Araçatuba, São Paulo, Brazil

* valeria.lima@unesp.br

## Abstract

Visceral leishmaniasis in humans is a chronic and fatal disease if left untreated. Canine leishmaniasis (CanL) is a severe public health problem because infected animals are powerful transmitters of the parasite to humans via phlebotomine vectors. Therefore, dogs are an essential target for control measures. Progression of canine infection is accompanied by failure of cellular immunity with reduction of circulating lymphocytes and increased cytokines that suppress macrophage function. Studies showed that the regulation of the effector function of macrophages and T cells appears to depend on miRNAs; miRNA-21 (miR-21) shows increased expression in splenic leukocytes of dogs with CanL and targets genes related to the immune response. Mimics and inhibitors of miR-21 were used *in vitro* to transfect splenic leukocytes from dogs with CanL. After transfection, expression levels of the proteins FAS, FASL, CD69, CCR7, TNF-α, IL-17, IFN-γ, and IL-10 were measured. FAS, FASL, CD69, and CCR7 expression levels decreased in splenic leukocytes from dogs with CanL. The miR-21 mimic decreased CD69 expression in splenic leukocytes from CanL and healthy groups. The miR-21 inhibitor decreased IL-10 levels in culture supernatants from splenic leukocytes in the CanL group. These findings suggest that miR-21 alters the immune response in CanL; therefore, miR-21 could be used as a possible therapeutic target for CanL.

## Introduction

Visceral leishmaniasis (VL), also known as *kala-azar*, is caused by the protozoan *Leishmania infantum (*syn *Leishmania chagasi)* and is fatal if left untreated in over 95% of cases. An estimated 50000 to 90000 new cases of VL occur worldwide annually, and most cases occur in Brazil, East Africa, and India [1]. Dogs are considered the primary domestic reservoirs of *L. infantum* [2]. In humans and dogs, the parasite causes characteristic symptoms of the disease [3,4]. The most frequent clinical signs of canine leishmaniasis (CanL) are lymphadenopathy, onychogryphosis, cutaneous lesions, weight loss, cachexia, fever, and locomotor abnormalities [5].

Dogs with CanL present a complex immune response, which is decisive for disease resistance or susceptibility [6,7]. The protective immune response against *Leishmania* sp. in dogs is

**Data Availability Statement:** All relevant data are within the manuscript and its Supporting Information files.

**Funding:** This work was supported by São Paulo Research Foundation (FAPESP) (www.fapesp.br), grant 2018/17261-5 and 2018/16239-6; National

Council for Scientific and Technological Development (CNPq), process 302165/2018-5 and 140460/2018-7. This study was also partially financed by the Coordination for the Improvement of Higher Education Personnel (CAPES) - Finance Code 001 (student supported: (GTR, JHF, MOS, SFC). The funders had no role in study design, data collection and analysis, decision to publish, or preparation of the manuscript. VMFL GRANT FAPESP 2018/17261-5 JPB SCHOLARSHIP FAPESP 2018/16239-6.

**Competing interests:** The authors have declared that no competing interests exist.

related to the increase in the Th1 response with the production of IFN-γ and IL-2, cytokines that promote activation of macrophages and cytotoxic T lymphocytes [8]. Susceptibility to infection has been associated with the predominance of IL-4 and IL-10; reducing the effects of Th1 cytokines that decrease the production of nitric oxide in macrophages prevents the destruction of the parasite [9].

MicroRNAs (miRNAs) are small non-coding RNAs that work as post-transcriptional regulators of gene expression, regulating the translation of proteins fundamental to the immune response [10]. In splenic leukocytes [11] and peripheral blood mononuclear cells from dogs with CanL [12], microarray analysis showed increased expression of miR-21 in the CanL group. The parasitic load in the spleens of these animals negatively correlated with the expression of miR-21 [11].

*In silico* analysis of pathways and targets of differentially expressed miRNAs in splenic leukocytes from dogs naturally infected by *L. infantum*, some genes related to the immune response in CanL were targets of miR-21 [11]. The canonical pathways included the miR-21 target genes FAS, FASL, CCR7, CD69, and TNF-α [11,12].

IL-10 is associated with susceptibility in CanL, and high levels of this cytokine are detected in the spleens of dogs with CanL [13]. There was a positive correlation between splenic levels of this cytokine and the progression of CanL [14]. In peripheral blood mononuclear cells of dogs with CanL, increased IL-10 levels were associated with the detection of parasitic DNA [15]. IL-10 mRNA was a target of miR-21 in a murine model of adenovirus [16] and autoimmune encephalomyelitis [17].

IFN-γ is a Th1 cytokine associated with resistance in CanL [13]. Significantly higher IFN-γ concentrations were noted in dogs in stage I of the disease [18]. IFN-γ-producing dogs presented lower antibody levels and lower blood parasitemia [18]. IL-17 is a mediator of inflammatory reactions in CanL, and the infection inhibited IL-17A mRNA expression in the spleen, especially in symptomatic dogs [19]. IL-17A acts synergistically with IFN-γ to promote protection against *L. infantum* infection [20]. IL-17 and IFN-γ mRNAs were targets of CD69 in murine CD4+ cells [21,22].

In the present study, we demonstrate that miR-21 overexpression leads to a decreased CD69 expression in splenic leukocytes of infected and healthy dogs, whereas decreased miR-21 expression leads to lower expression of IL-10 in culture supernatants from dogs with CanL.

## Materials and methods

### Animal screening and collection of samples

The Committee for Ethics in Animal Experimental Research approved the study, with the approval of the Committee for Ethics in Animal Use of São Paulo State University, School of Veterinary Medicine, Araçatuba (process number 00624–2018). The owners of the control group dogs did the consent to the surgery by written the term of consentient, according Committee for Ethics instructions.

Five healthy dogs were used in the control group. These animals were selected following clinical examination, complete blood count, and serum biochemical profile within the normal range for the species and negative results for CanL (serological [23] and molecular [24]; Table 1). Dogs selected are of both sexes and have between 1–5 years old.

In the infected group, ten dogs were naturally infected with *L. infantum*; all animals were positive for leishmaniasis by serology and molecular testing and were kept at the Zoonosis Control Center of Araçatuba. These animals carried at least three characteristic clinical signs of the disease, including onychogryphosis, weight loss, ear-tip injuries, periocular lesions, alopecia, skin lesions, or lymphadenopathy (Table 1).

**Table 1. Screening of dogs.** Optical density on ELISA and clinical signs of CanL and control groups.

| Animal | O.D. (ELISA) | Sex | Clinical Signs | PCR |
|---|---|---|---|---|
| Infected 1 | 1,132 | F | Onychogryphosis, skin lesions, cachexia, seborrhea | + |
| Infected 2 | 1,065 | M | Lymphadenopathy, onychogryphosis, cachexia and skin lesions | + |
| Infected 3 | 0,978 | F | Lymphadenopathy, onychogryphosis, ear injuries, alopecia, skin lesions | + |
| Infected 4 | 0,636 | M | Lymphadenopathy, cachexia, skin lesions, periocular lesion | + |
| Infected 5 | 1,374 | F | Onychogryphosis, cachexia, alopecia, skin lesions | + |
| Infected 6 | 1,208 | M | Lymphadenopathy, onychogryphosis, cachexia, alopecia | + |
| Infected 7 | 1,267 | F | Lymphadenopathy, onychogryphosis, seborrhea, alopecia, skin lesions, periocular lesion, hepatosplenomegaly | + |
| Infected 8 | 1,052 | F | Lymphadenopathy, onychogryphosis, periocular lesion, hepatosplenomegaly | + |
| Infected 9 | 0,968 | F | Lymphadenopathy, onychogryphosis, cachexia | + |
| Infected 10 | 1,049 | F | Lymphadenopathy, onychogryphosis, seborrhea | + |
| Control 1 | 0,062 | F | No clinical signs | - |
| Control 2 | 0,026 | M | No clinical signs | - |
| Control 3 | 0,147 | F | No clinical signs | - |
| Control 4 | 0,028 | M | No clinical signs | - |
| Control 5 | 0,071 | F | No clinical signs | - |

Blood samples from the healthy and infected groups were collected in tubes without EDTA to obtain serum for biochemical profiles (S1 Table) and indirect ELISA (Table 1) to measure anti-leishmanial antibodies [23]. Blood was collected in EDTA-containing tubes for complete blood count (S2 Table). Infected dogs were euthanized by barbiturate anesthesia (Tiopental, Cristália Itapira, SP), followed by intravenous injection of 19.1% potassium chloride, as recommended for VL control in compliance with local legislation. After euthanasia, a 2-cm$^3$ fragment of the spleen was collected for isolation of splenic leukocytes. Splenic fragments in control dogs were removed by surgical excision as described by [25]. Dogs with laboratory exams incompatible with clinical leishmaniasis were not used in the study.

## Isolation of splenic leukocytes

Splenic leukocytes were obtained from a 2-cm$^3$ fragment that was macerated using a mortar and pestle, and added to 10 ml RPMI-1640 medium (Sigma, USA) supplemented with 10% heat-inactivated fetal bovine serum (FBS), 0.03% L-glutamine, and 100 IU/mL penicillin and 100 mg/mL streptomycin. After removal of cell debris through a 100-μm cell strainer (BD Falcon Cell strainer, USA), suspensions were processed with 5 mL of red blood cell lysis buffer containing 7.46 g/L ammonium chloride ($NH_4ClO_3$), 1,6 g/L EDTA and 0,84 g/L sodium carbonate ($Na_2CO_3$) at 4° C for 10 minutes, centrifuged at 2000 rpm for 5 minutes, and washed with phosphate-buffered saline (PBS) at pH 7.2 three times. Cells were counted in a Neubauer chamber.

## Serological diagnosis by ELISA

Samples were analyzed by ELISA using total antigen from lysed promastigotes [26]. The antigen was coated overnight with 20 μg/ml protein pH 9.6, then washed three times in PBS containing 0.05% Tween 20 (washing buffer) and saturated for 1 hour with 150 μl/well of a mixture of PBS and 10% FBS at room temperature. Next, the preparation was washed three times with washing buffer. Blocking buffer/Tween (100 μl of serum sample (1/400) diluted in PBS, pH 7.2, containing 0.05% Tween 20 and 10% FCS) was added to each well and incubated at room temperature for 3 h, followed by three washes with washing buffer. Subsequently, 100 μl/well of anti-dog IgG conjugated with horseradish peroxidase (Sigma, St. Louis, MO, USA) at appropriate dilution in

blocking buffer/Tween was added, incubated at room temperature for 1 hour, and washed. Substrate solution (0.4 mg/ml o-phenylenediamine (Sigma) and 0.4 μl/ml $H_2O_2$ in phosphate citrate buffer, pH 5.0) was added at 100 μl/well and developed for 5 min at room temperature. The reaction was stopped with 50 μl of 3M $H_2SO_4$. Absorbance was measured at 490 nm using a Tecan microplate reader (Sunrise model ref. 16039400). Negative and positive controls were included on each plate. Positive controls obtained from a hyperimmune animal were included. The cut-off was determined using the mean +3 SD of the readings obtained from serum samples of healthy dogs from non-endemic areas for leishmaniasis.

## DNA extraction and determination of the *Leishmania* species

DNA extraction from splenic leukocytes samples from the experimental dogs was performed using $5 \times 10^6$ cells with the commercial DNAeasy kit (Qiagen, USA) according to the manufacturer's recommendations. Extracted DNA was quantified in a spectrophotometer 260/280 (NanoDrop, Thermo Fisher Scientific) to measure purity and concentration and were then stored at –20˚C until analysis.

Determination of the *Leishmania* species was performed by polymerase chain reaction (PCR)-restriction fragment length polymorphism [24], comparing the restriction profiles of the sample with a PCR restriction profile obtained from *L. infantum* (IOC / L0575-MHOM / BR / 2002 / LPC-RPV), *L. braziliensis* (IOC / L0566-MHOM / BR / 1975 / M2903) and *L. amazonensis* (IOC / L0575-MHOM / BR / 1967 / PH8) as positive controls, and water as a negative control (S1 Fig).

## Extraction and quantification of total RNA

Extraction of total RNA, including miRNAs, from $5 \times 10^4$ splenic leukocytes post-transfection, was performed using the commercial mirVana kit for isolation of total RNA with phenol (Life Technologies, USA), following manufacturer's instructions. After RNA isolation, samples were stored at –80˚C.

RNA samples were analyzed in a spectrophotometer (NanoDrop, Thermo Scientific, USA) for purity evaluation (260/280) and quantification.

## Real-time PCR for miR-21

To confirm that miR-21 is upregulated in dogs with CanL obtained by [11], real-time quantitative PCR (qPCR) was performed. cDNA production was performed using the miScript RT II kit (Qiagen, USA), as recommended by the manufacturer. A total of 1 μg of RNA was used for each sample with the 5x miScript Hiflex Buffer, in a final volume of 20 μl. Mix was incubated for 60 min at 37˚C, followed by 5 min at 95˚C to inactivate the miScript Reverse Transcriptase. Next, qPCR was performed using commercially available specific primer for *Canis familiaris* miR-21 and the endogenous reference RNA SNORD96A, as recommended by manufacturer (miScript, Qiagen). The SYBR Green system (MyScript SYBR Green PCR Kit, Qiagen) was used in a real-time thermal cycler (RealPlex, Eppendorf). Amplification conditions consisted of an initial activation step of 95˚C for 15 min followed by 40 cycles of 94˚C for 15 seconds, 55˚C for 30 seconds, and 70˚C for 30 seconds denaturation, annealing, and extension, respectively. For miRNA analysis, a standard curve was generated with serial dilution of a pool of all cDNAs. The absolute quantification of miR-21 was performed by converting the sample cycle threshold values to a concentration (ng/μl) based on the standard curves generated using 10-fold serial dilutions of the cDNA pool. Values obtained for the target miRNA were then divided by SNORD96A values to obtain normalized target values for each sample. All samples were run in duplicate.

## Transfection with miR-21 mimic and inhibitor in splenic leukocytes

Splenic leukocytes were cultured (1.6 x $10^5$ cells/replicate) in triplicate in 24-well plates for 48 h at 37°C in 5% $CO_2$. All-Stars Negative control siRNA (scrambled), miR-21 mimic (5 nM), and miR-21 inhibitor (50 nM) (miScript miRNA Mimic and Inhibitor Qiagen, USA) were used, and splenic leukocytes were transfected using 3 μL of Hiperfect (Qiagen, USA) in each well, following manufacturer's instructions. To evaluate transfection rates, AllStars HS Cell Death Control siRNA (Qiagen, USA) was used at a final concentration of 50 nM. AllStars Hs Cell Death Control siRNA is a blend of highly potent siRNAs targeting ubiquitously expressed genes that are essential for cell survival. Knockdown of these genes induces a high degree of cell death. The transfection rate was measured by flow cytometry using 7-AAD Viability Staining Solution (BioLegend, USA) according to the manufacturer's instructions. Cell death was evaluated using Trypan blue in a Neubauer chamber for optical microscopy. A medium transfection rate of 20% was obtained for both groups. Experiment to confirm the transfection rate are demonstrate in representative image (S2 Fig).

## Flow cytometry analysis in splenic leukocytes

For flow cytometry analysis, 1 x $10^4$ cells were incubated with Fc blocking buffer (10% FBS) for 30 min at room temperature. Cells were centrifuged at 1800 rpm for 7 minutes and then incubated with phycoerythrin (PE)-conjugated anti-human CD95 (FAS) monoclonal antibody (BD Biosciences, USA), anti-human CD178 (FASL) monoclonal antibody (BD Biosciences, USA), and anti-human CD69 polyclonal antibody (Lifespan Biosciences, USA). To measure CCR7 in dendritic cells, splenic leukocytes were incubated with PE-conjugated anti-human CCR7 monoclonal antibody (Invitrogen, USA), anti-dog MHC class II conjugated with fluorescein isothiocyanate (FITC) (Bio-Rad, USA), or anti-human CD11c conjugated with peridinin-chlorophyll-protein (Lifespan Biosciences, USA). To avoid non-specific binding, cells were incubated with respective control isotypes. Acquisition of 10,000 events was counted by experimental replicate on channels FL1, FL2, and FL3, and cytometric analysis was performed using an Accuri C5 Flow Cytometer (BD Biosciences, USA) with BD Accuri C6 software, version 1.0.264.21 (BD Biosciences, USA).

To determine CD69 expression on lymphocytes, splenic leukocytes were incubated with Fc blocking buffer (10% FBS) for 30 min at room temperature. Cells were centrifuged at 1800 rpm for 7 minutes and then incubated with anti-dog CD4 monoclonal antibody (FITC) (ABD Serotec, USA), anti-dog CD8 monoclonal antibody (FITC) (ABD Serotec, USA), or anti-human CD21 monoclonal antibody (FITC) (ExBio, Czech Republic) in different tubes, and with PE-conjugated anti-human CD69 polyclonal antibody (Lifespan Biosciences, USA) to obtain double tagging of CD69 in T CD4+, T CD8+, and B lymphocytes, respectively. To avoid non-specific binding, cells were incubated with respective control isotypes. Acquisition of 10,000 events was counted by experimental replicate on channels FL1 and FL2, and cytometry was performed using an Accuri C5 Flow Cytometer (BD Biosciences, USA) using BD Accuri C6 software version 1.0.264.21 (BD Biosciences, USA).

## Dosage of cytokines by ELISA

After 48 h of transfection, supernatants from splenic leukocyte cultures were collected, centrifuged at 2500 rpm, and stored at −80°C until further analysis. Concentrations of TNF-α [12], IL-10 [17], IL-17, and IFN-γ [21] in the supernatant were determined by capture ELISA using a Canine DuoSet ELISA Kit (R&D Systems, USA) for the respective cytokines. The assay was performed according to the manufacturer's instructions. The plates were read using a Spectra Count™ reader (Packard BioScience Company) with a 450-nm filter. All measurements were performed in duplicate.

### Statistical analysis

Statistical analysis was performed using GraphPad Prism 6 software (GraphPad Software, Inc., La Jolla, CA, USA). D'Agostino & Pearson, Shapiro–Wilk, and Kolmogorov-Smirnov tests were determined to assess for normality of distribution, and then non-parametric tests were used. The Mann–Whitney test was used for group comparison. Treatment comparisons (miR-21 mimic, miR-21 inhibitor, scrambled, hiperfect and untransfected cells) were evaluated using the Friedman test followed by Dunn's multiple comparisons test (comparing the mean rank of each treatment with every other treatments). Differences were considered significant when $p < 0.05$.

## Results

### miR-21 expression is increased in CanL

Because miR-21 regulates the immune response, to confirm the increase in miR-21 expression in CanL, real-time PCR was performed with samples of splenic leukocytes from dogs of both groups. We found a higher expression of miR-21 in dogs with CanL than healthy dogs (Fig 1).

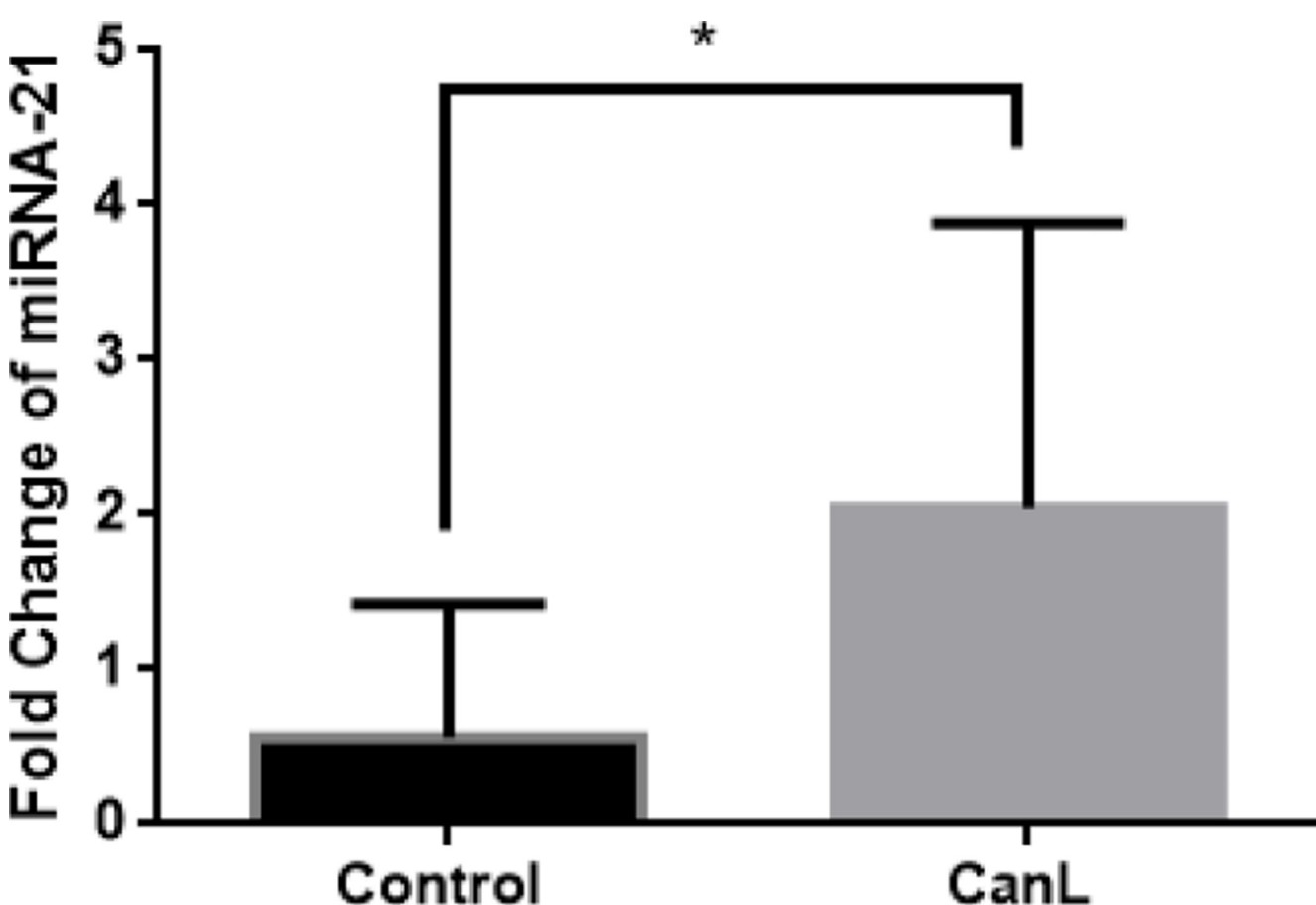

**Fig 1. Expression of miR-21.** miR-21 expression using real-time PCR. Expression of miR-21 was quantified using real-time PCR in splenic leukocytes of CanL (n = 10) and control (n = 5) dogs. Fold change of miR-21 was calculated with normalized results by converting the sample cycle threshold values to a concentration (ng/μl). Data represent the mean values of miRNA expression ± standard deviation, and the asterisks represent statistically significant data following the Mann–Whitney test. Results were considered significant when p < 0.05.

### Expression of proteins regulated by miR-21 in splenic leukocytes of CanL and healthy dogs

Expression of FAS, FASL, CD69, and CCR7 were compared between control and CanL groups. Proteins FAS (Fig 2A), FASL (Fig 2B), CD69 (Fig 2C), and CCR7 (Fig 2D), regulated by miR-21 [11,12] were decreased in splenic leukocytes from the CanL group.

### miR-21 mimics lead to decreased CD69 expression by B lymphocytes in healthy dogs and those with CanL

To determine whether miR-21 affects the expression of FAS, FASL, CD69, and CCR7, splenic leukocytes were transfected with miR-21 mimics and inhibitors, and after 48 hours, protein expression was measured using flow cytometry. Expression of FAS, FASL, and CCR7 presented no statistically significant difference after transfection with mimics and inhibitors of miR-21 in CanL and healthy groups (S3 Fig). A decrease in CD69 protein expression was found in splenic leukocytes after transfection with miR-21 mimic in the control group (Fig 3A) and the CanL group (Fig 3B).

To determine which cell subpopulation showed decreased CD69 expression, splenic leukocytes from the CanL group were transfected with miR-21 mimics and inhibitors, and CD69 expression in CD4+, CD8+, and CD21+ cells was obtained by double tagging (S4 Table). We found significantly lower CD69 expression only in B lymphocytes (CD21+) after transfection with miR-21 mimics compared with scrambled (Fig 4).

### IL-10 expression decreased in the presence of miR-21 inhibitor

To determine whether miR-21 regulates the expression of cytokines IL-10, TNF-α, IFN-γ, and IL-17, splenic leukocytes from the CanL group were transfected with miR-21 mimics and inhibitors, and after 48 hours, cytokine concentrations were measured by capture ELISA in cell culture supernatants. We observed a decrease in IL-10 in culture supernatants in the presence of miR-21 inhibitor in the CanL group (Fig 5). TNF-α, IFN-γ, and IL-17 showed no significant differences (S4 Fig).

## Discussion

Dogs naturally infected with *L. infantum* that develop disease show an inability to mount a specific effective adaptive immune response; miRNAs, including miR-21, could be responsible for modulation of the immune system. To elucidate the role of miR-21 in dogs with CanL, we evaluated targets of miR-21 after transfection with mimics and inhibitors. Real-time PCR confirmed higher expression of miR-21 in dogs with CanL than healthy dogs. *In silico* analysis showed that miR-21 targets the genes FAS, FASL, CCR7, CD69, TNF-α, and IL-10. Expression levels of proteins FAS, FASL, CCR7, and CD69 were decreased in dogs with CanL. Next, the role of miR-21 was evaluated using splenic leukocyte transfection with miR-21 mimics and inhibitors, and protein expression was studied. We found that FAS, FASL, and CCR7 expression was not regulated by miR-21; however, the miR-21mimic decreased CD69 expression in B lymphocytes and the inhibition of miR-21 decreased IL-10 in culture supernatants from splenic leukocytes.

We observed lower expression levels of CD95 (FAS) and CD178 (FASL) in splenic leukocytes from infected dogs than dogs in the control group. FAS and FASL play critical roles in the immune system, particularly in the death of target cells infected by pathogens and obsolete and potentially dangerous lymphocytes [27]. FASL-FAS signaling triggers apoptosis through recruitment mediated by FADD adapter proteins (FAS-associated protein with death domain,

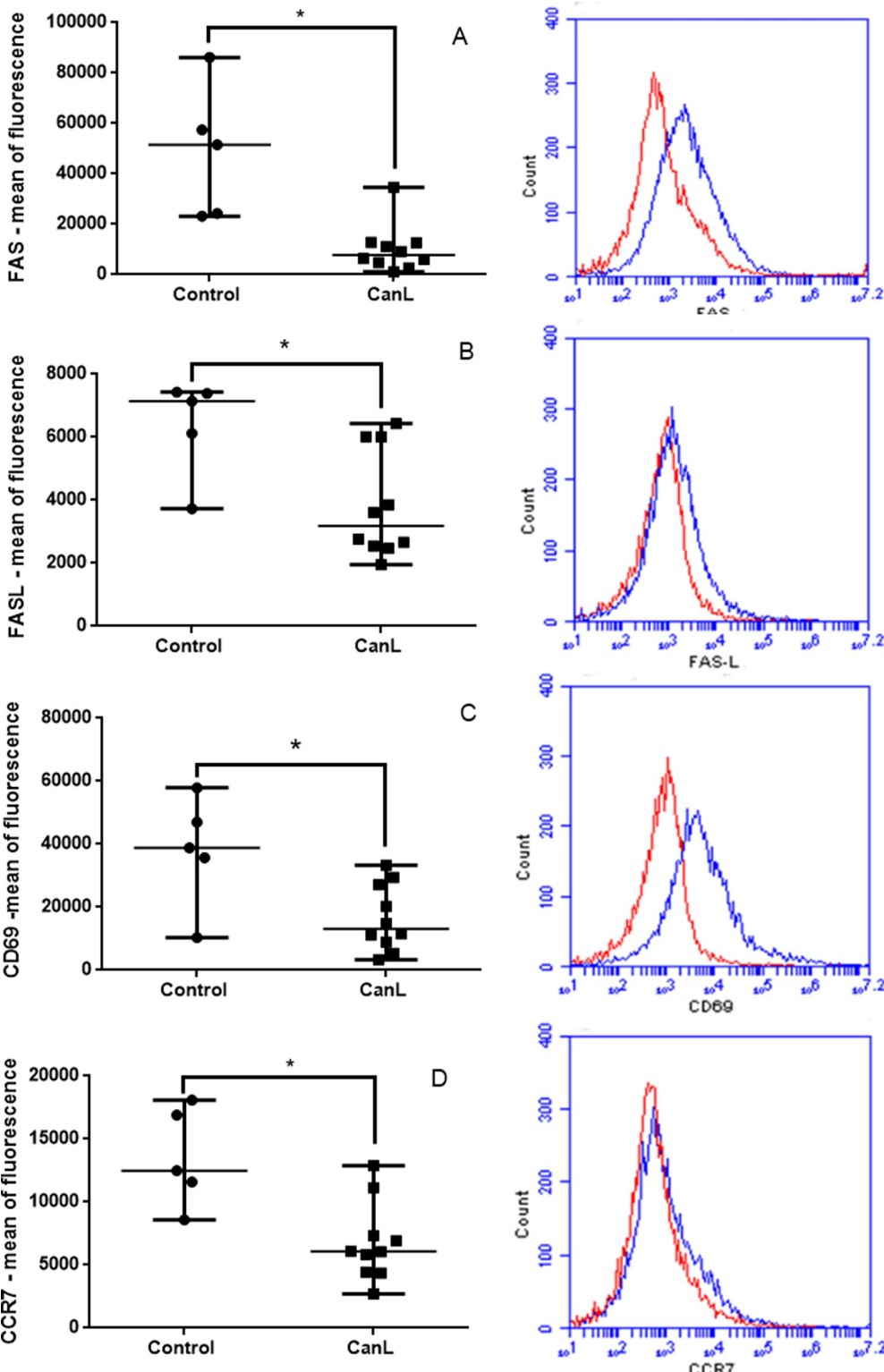

**Fig 2. Expression of proteins regulated by miR-21 in splenic leukocytes of CanL and healthy dogs.** Expression of FAS (A), FASL (B), CD69 (C), and CCR7 (D) in splenic leukocytes from CanL and healthy dogs after culture and respective representative histograms obtained from flow cytometry analysis. The red line represents the CanL group, and the blue line represents the control group. Cells were cultured for 48 h at 37°C and 5% $CO_2$ without treatment (medium) and then incubated with monoclonal antibodies. Data are presented as median ± min-max. Asterisks represent statistical significance (Mann–Whitney, *p < 0.05).

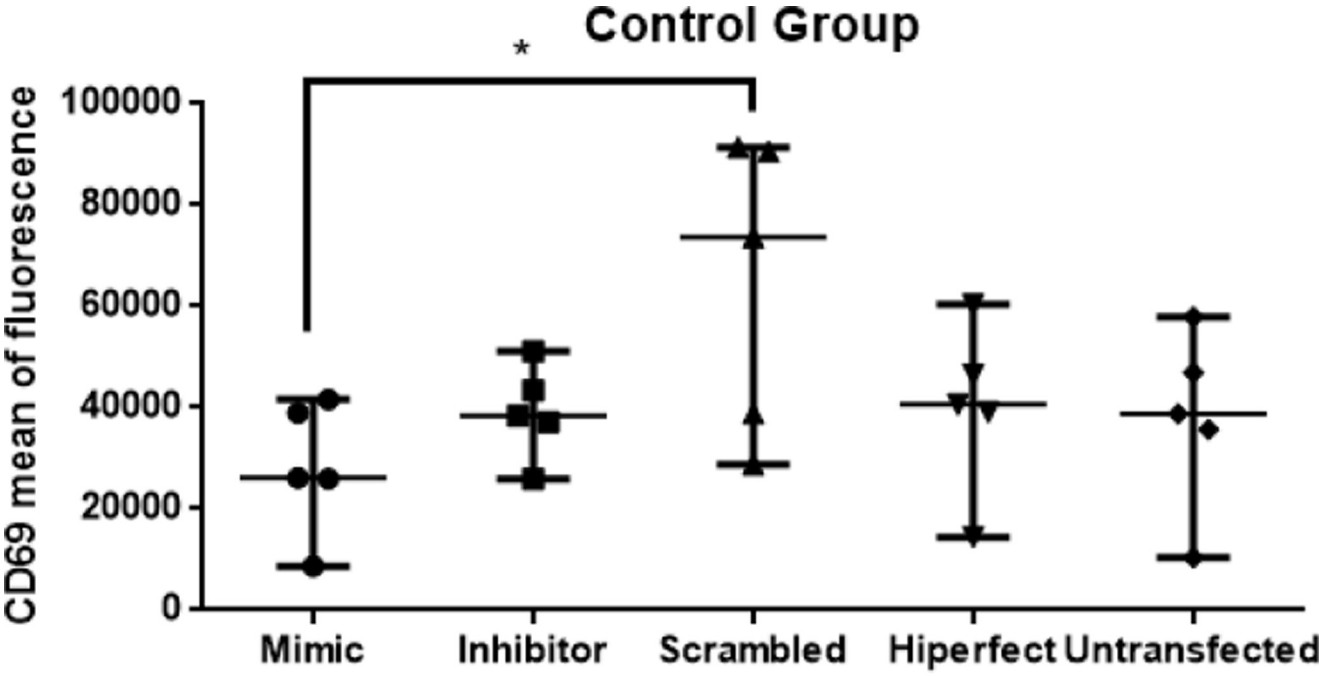

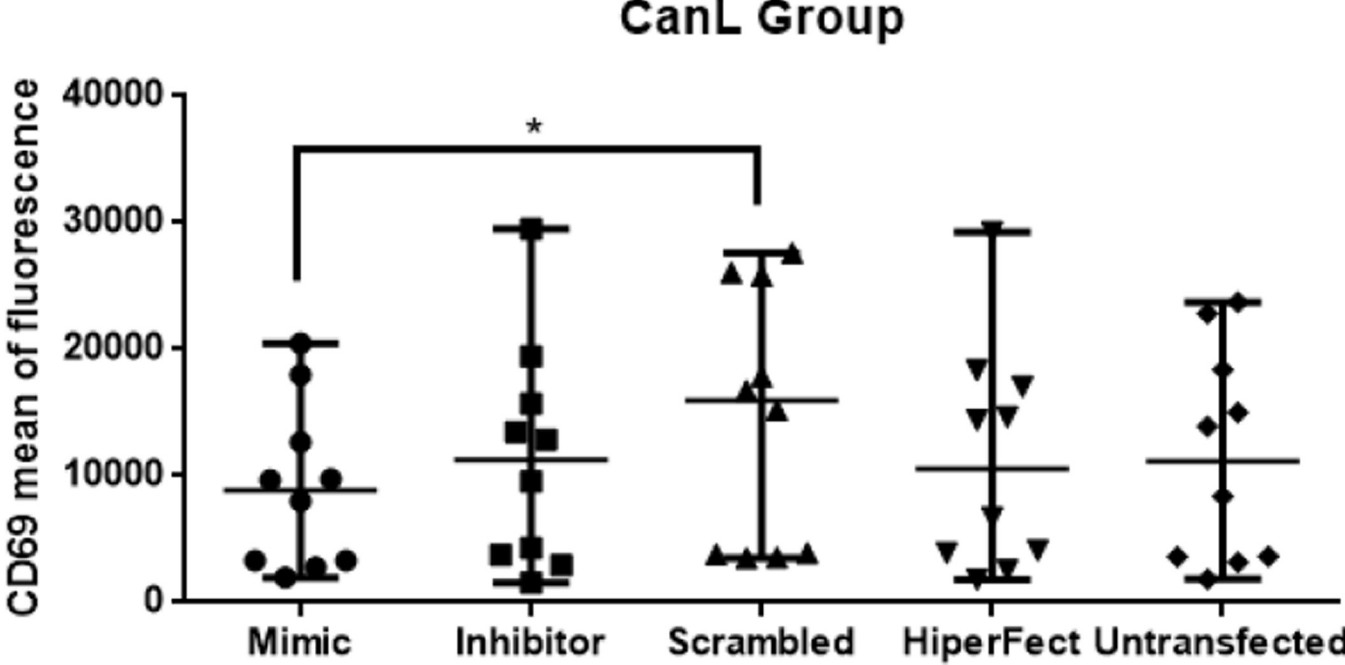

**Fig 3. Expression of CD69 after transfection of splenic leukocytes with mimics and inhibitors of miR-21.** Expression of CD69 protein in splenic leukocytes of the control (A) and CanL group (B). Splenic leukocytes of dogs naturally infected by *L. infantum* and healthy dogs were transfected with scrambled, miR-21 mimic, and miR-21 inhibitor, all in the presence of Hiperfect, following 48 h in culture at 37˚C and 5% $CO_2$. Data are presented as median ± min-max. The asterisk indicates significant differences (Friedman's multiple comparison test followed by Dunn's multiple comparisons test (comparing the mean rank of each treatment with every other treatments), * $p < 0.05$).

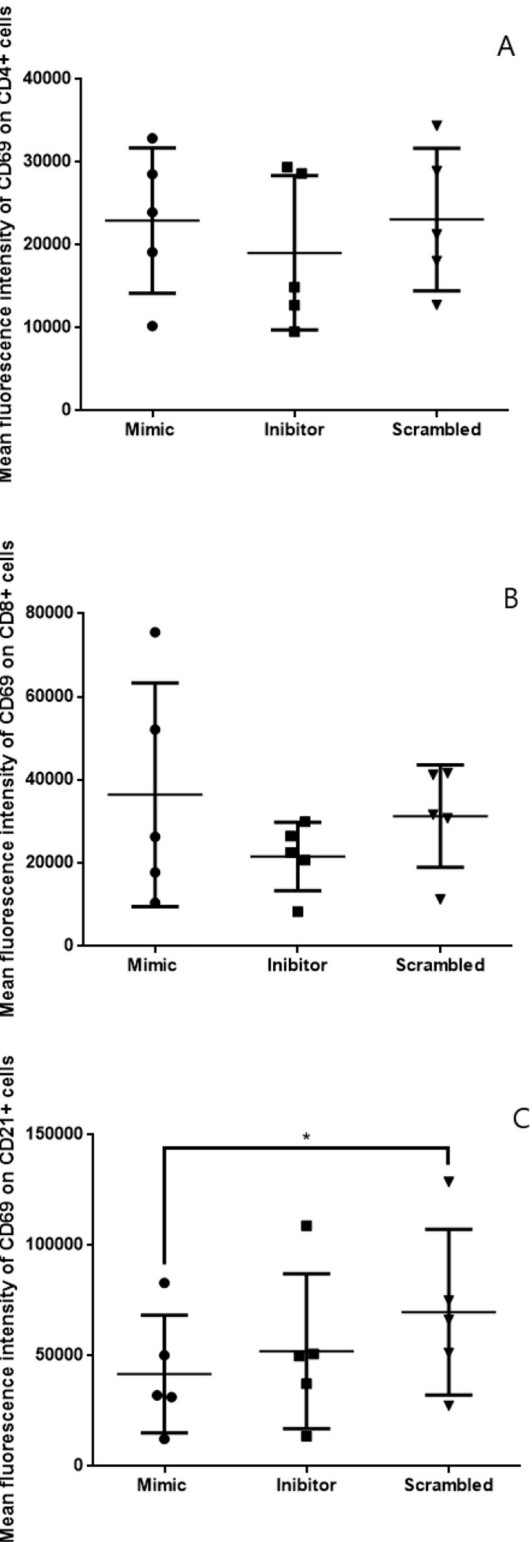

**Fig 4. CD69 expression in T CD4+, T CD8+ and B lymphocytes.** Expression of CD69 protein in (A) CD4+, (B) CD8 + and (C) B lymphocytes (CD21+ cells) in the CanL group. Splenic leukocytes of dogs naturally infected by *L. infantum* were transfected with scrambled, miR-21 mimic, and miR-21 inhibitor, all in the presence of Hiperfect, following 48 h in culture at 37°C and 5% $CO_2$. Data are presented as median ± min-max, and the asterisk indicates significant differences (Friedman's multiple comparison test, $^*$ $p < 0.05$).

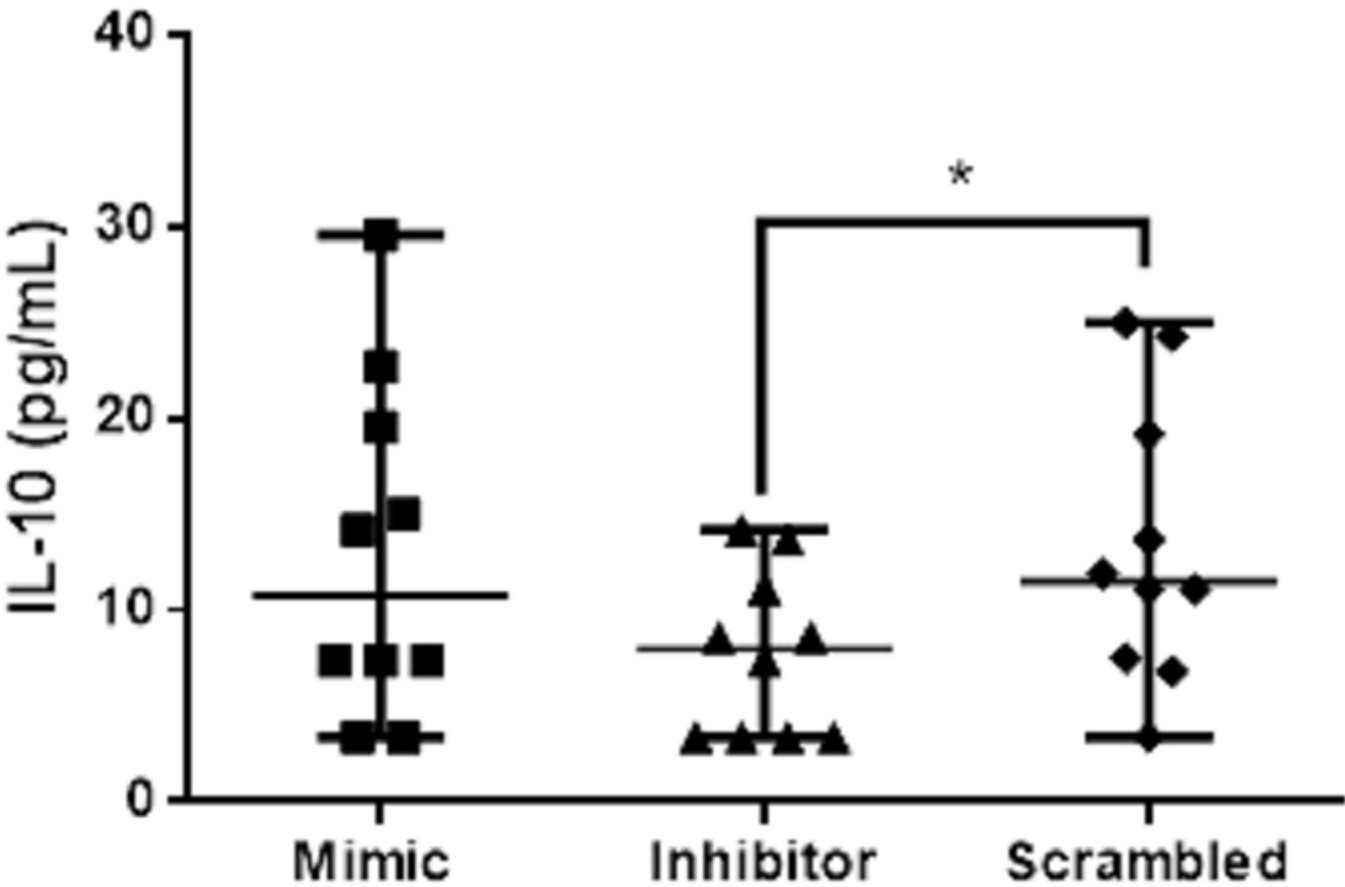

**Fig 5. Concentration of IL-10 after transfection of splenic leukocytes with mimics and inhibitors of miR-21.** IL-10 production was quantified in supernatants from splenic leukocytes cultures of dogs naturally infected by *L. infantum* and transfected with scrambled, miR-21 mimic, and miR-21 inhibitor, all in the presence of Hiperfect, following 48 h in culture. Data represent the median values of the IL-10 + min-max. Asterisks represent significance (p < 0.05) by the Friedman Test with the Dunn multiple comparisons.

also called MORT1) and caspase-8 activation [27]. Similar to our finding, low FAS and FASL expression levels were observed in spleen and peripheral blood CD4+ cells from dogs with CanL [28]. The increase or decrease in miR-21 in cultures did not alter the expression of FAS and FASL, suggesting that, in dogs, the mRNA of these proteins may not be targets of miR-21. The low transfection rate obtained may not have been sufficient to modulate the expression of these molecules.

CCR7 protein showed lower expression in splenic leukocytes from the CanL group than the control group. CCR7, a chemokine receptor important for cell migration, is expressed in dendritic cells and its mRNA is a target of miR-21 in human CD4+ T cells [29]. Dendritic cells from mice infected with *L. donovani* showed impairment of migration from the marginal zone to the periarteriolar region of the spleen, partly attributed to the inhibition of CCR7 expression [30]. The interaction between T cells and dendritic cells is essential to generate an adaptive immune response; mature dendritic cells from mice with CCR7 deficiency do not migrate to draining lymph nodes after activation, making it impossible to mount a rapid response of primary B or T cells. [31]. These findings suggest that it is possible in CanL that the low expression of CCR7 compromises cell migration; nevertheless, we did not observe regulation of the expression of CCR7 by miR-21, which may be due to low transfection rates.

We also found lower expression of CD69 in splenic leukocytes from dogs with CanL than healthy dogs. CD69 gene is a target of miR-21 in splenic leukocytes of dogs with CanL. CD69 is a cell surface molecule and is one of the first to be expressed after activation of T and B lymphocytes and other cells of hematopoietic origin [32,33]. CD69 expression rapidly increases after activation in most leukocytes, highlighting its widespread use as a marker of activated lymphocytes and NK cells [21]. In addition to its intrinsic value as an activation marker, CD69 is also an essential regulator of immune responses [21]. Therefore, it is crucial to elucidate the role of CD69 expression in the function of immune cells in *L. infantum* infection of dogs.

A decrease in CD69 expression was observed in splenic leukocytes in the presence of miR-21 mimic compared to the negative control (scrambled), both in dogs with CanL and in healthy dogs. The low expression of CD69 observed in dogs with CanL contrasts with results in experimental models in mice infected with *L. infantum*, where an increase in the percentage of CD69+ cells was observed in the spleen in the acute phase [34] and in the chronic phase of the disease where activation of protective immunity reduces the splenic parasitic load [35]; this is not seen in CanL, where the disease is progressive. These results suggest that miR-21 reduces the expression of CD69 by B cells, possibly regulating B lymphocyte function.

There was a decrease in the expression of IL-10 in the presence of miR-21 inhibitor in the CanL group. IL-10 is the primary cytokine suppressing the immune response in humans and a murine model of VL [36,37]. In CanL, increased levels of IL-10 were described [38], and this increase was associated with the detection of parasitic DNA [15], confirming the regulatory role of IL-10 in the spleen. Differently to our finding, in naive T cells from healthy humans, the induction of miR-21 led to an increase in IL-10 expression [39]. These findings suggest that the parasite may be using IL-10 as an escape mechanism, modulating immune responses through miR-21.

We conclude that *L. infantum* infection in dogs increases expression miR-21 that regulates CD69 and IL-10 expression, essential proteins involved in the immune response to the parasite.

## Supporting information

**S1 Fig. PCR-RFLP.** Restriction fragment length polymorphism analysis of ITS1-PCR fragments amplified from DNA samples using Hae III enzyme. NC: Negative control (water); M: molecular marker (123 bp); La: *Leishmania amazonensis* (IOC / L0575-MHOM / BR / 1967 / PH8); Lb: *Leishmania braziliensis* (IOC / L0566-MHOM / BR / 1975 / M2903); Li: *Leishmania infantum* (IOC / L0575-MHOM / BR / 2002 / LPC-RPV); C1 to C5: control group; CanL 1 to CanL10: CanL group. CanL sample profiles were identical to *L. infantum*.
(TIFF)

**S2 Fig. Control of transfection.** Representative histogram obtained from flow cytometry analysis. The red line represents the cells cultured with reagent Cell Death, and the black line represents the cells cultures without any transfection reagent. Cells were cultured for 48 h at 37°C and 5% $CO_2$.
(TIFF)

**S3 Fig. Expression of FAS, FASL and CCR7 after transfection of splenic leukocytes with mimics and inhibitors of miR-21.** Expression of FAS (A), FASL (B) and CCR7 (C) proteins in splenic leukocytes of the CanL and Control groups. Splenic leukocytes of dogs naturally infected by L. infantum and healthy dogs were transfected with scrambled, miR-21 mimic, and miR-21 inhibitor, all in the presence of Hiperfect, following 48 h in culture at 37°C and 5% CO2. Data are presented as median ± min-max. The asterisk indicates significant differences

(Friedman's multiple comparison test, $^*$ $p < 0.05$).
(TIFF)

**S4 Fig. Concentration of IL-10, TNF-α, IFN-γ, and IL-17 after transfection of splenic leukocytes with mimics and inhibitors of miR-21.** Splenic leukocytes from the CanL group were transfected with miR-21 mimics and inhibitors, and after 48 hours, cytokine concentrations were measured by capture ELISA in cell culture supernatants. Data represent the median values + min-max. Asterisks represent significance ($p < 0.05$) by the Friedman Test with the Dunn multiple comparisons.
(TIFF)

**S1 Table. Biochemical profiles of CanL and control groups.**
(DOCX)

**S2 Table. Complete blood counts of CanL and control groups.**
(DOCX)

**S3 Table. CD69 expression in lymphocytes.**
(DOCX)

**S4 Table. CD69 expression in lymphocytes.** Mean±SD of CD69 in CD4+, CD8+ and CD21 + cells measured by flow cytometry in splenic leukocytes of CanL group after transfection with miR-21 mimics and inhibitors for 48h at 37˚C and 5% $CO_2$.
(DOCX)

**S1 Raw images.**
(PDF)

## Author Contributions

**Conceptualization:** Jaqueline Poleto Bragato, Valéria Marçal Felix de Lima.

**Data curation:** Jaqueline Poleto Bragato, Valéria Marçal Felix de Lima.

**Formal analysis:** Jaqueline Poleto Bragato, Valéria Marçal Felix de Lima.

**Funding acquisition:** Valéria Marçal Felix de Lima.

**Investigation:** Jaqueline Poleto Bragato, Valéria Marçal Felix de Lima.

**Methodology:** Jaqueline Poleto Bragato, Gabriela Torres Rebech, Jéssica Henrique de Freitas, Marilene Oliveira dos Santos, Sidnei Ferro Costa, Flavia de Rezende Eugênio, Paulo Sérgio Patto dos Santos, Valéria Marçal Felix de Lima.

**Project administration:** Valéria Marçal Felix de Lima.

**Resources:** Jaqueline Poleto Bragato, Valéria Marçal Felix de Lima.

**Software:** Jaqueline Poleto Bragato.

**Supervision:** Valéria Marçal Felix de Lima.

**Validation:** Jaqueline Poleto Bragato.

**Visualization:** Jaqueline Poleto Bragato.

**Writing – original draft:** Jaqueline Poleto Bragato.

**Writing – review & editing:** Jaqueline Poleto Bragato, Valéria Marçal Felix de Lima.

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
