## [Decision Letter · Decision Letter 0]

3 Sep 2021

PONE-D-21-21939

miRNA-21 regulates CD69 and IL-10 expression in canine leishmaniasis

PLOS ONE

Dear Dr. de Lima,

Thank you for submitting your manuscript to PLOS ONE. After careful consideration, we feel that it has merit but does not fully meet PLOS ONE’s publication criteria as it currently stands. Therefore, we invite you to submit a revised version of the manuscript that addresses the points raised during the review process.

1) The authors should provide experiments and controls for the transfection efficacy of the micro RNA to be sure that the effect was due to the action or not of the transfected micro RNA;

2) Please, see the comments raised by the both reviewers.

We look forward to receiving your revised manuscript.

Kind regards,

Paulo Lee Ho, Ph.D.

Academic Editor

PLOS ONE

Journal Requirements:

2. In your Methods section, please provide additional details regarding participant consent from the owners of the animals. In the ethics statement in the Methods and online submission information, please ensure that you have specified (1) whether consent was informed and (2) what type you obtained (for instance, written or verbal). If the need for consent was waived by the ethics committee, please include this information

3. In your Methods section, please provide additional details regarding the dogs used in your study and ensure you have described the source. For more information regarding PLOS' policy on materials sharing and reporting, see https://journals.plos.org/plosone/s/materials-and-software-sharing#loc-sharing-materials

In your Funding Information please provide the grant number for:

coordenação de aperfeiçoamento de pessoal de nível superior 

http://dx.doi.org/10.13039/501100002322

Valéria Marçal Felix de Lima

5. We note that you have stated that you will provide repository information for your data at acceptance. Should your manuscript be accepted for publication, we will hold it until you provide the relevant accession numbers or DOIs necessary to access your data. If you wish to make changes to your Data Availability statement, please describe these changes in your cover letter and we will update your Data Availability statement to reflect the information you provide

7. We note that you have included the phrase “data not shown” in your manuscript. Unfortunately, this does not meet our data sharing requirements. PLOS does not permit references to inaccessible data. We require that authors provide all relevant data within the paper, Supporting Information files, or in an acceptable, public repository. Please add a citation to support this phrase or upload the data that corresponds with these findings to a stable repository (such as Figshare or Dryad) and provide and URLs, DOIs, or accession numbers that may be used to access these data. Or, if the data are not a core part of the research being presented in your study, we ask that you remove the phrase that refers to these data

8. PLOS ONE now requires that authors provide the original uncropped and unadjusted images underlying all blot or gel results reported in a submission’s figures or Supporting Information files. This policy and the journal’s other requirements for blot/gel reporting and figure preparation are described in detail at https://journals.plos.org/plosone/s/figures#loc-blot-and-gel-reporting-requirements and https://journals.plos.org/plosone/s/figures#loc-preparing-figures-from-image-files. When you submit your revised manuscript, please ensure that your figures adhere fully to these guidelines and provide the original underlying images for all blot or gel data reported in your submission. See the following link for instructions on providing the original image data: https://journals.plos.org/plosone/s/figures#loc-original-images-for-blots-and-gels. 

In your cover letter, please note whether your blot/gel image data are in Supporting Information or posted at a public data repository, provide the repository URL if relevant, and provide specific details as to which raw blot/gel images, if any, are not available. Email us at plosone@plos.org if you have any questions

Reviewers' comments:

Reviewer's Responses to Questions

**Comments to the Author**

1. Is the manuscript technically sound, and do the data support the conclusions?

Reviewer #1: Yes

Reviewer #2: Partly

2. Has the statistical analysis been performed appropriately and rigorously? 

Reviewer #1: Yes

Reviewer #2: Yes

3. Have the authors made all data underlying the findings in their manuscript fully available?

Reviewer #1: Yes

Reviewer #2: Yes

4. Is the manuscript presented in an intelligible fashion and written in standard English?

Reviewer #1: Yes

Reviewer #2: Yes

5. Review Comments to the Author

Reviewer #1: The paper entitles “miRNA-21 regulates CD69 and IL-10 expression in canine leishmaniasis” by de Lima et al., submitted for consideration in PLOS ONE journal (PONE-D-21-21939), shows interesting results in naturally Leishmania infected dogs regarding immune regulation by miRNA 21. However, some critical aspects should be addressed, particularly concerning the limitation of the work.

The paper was easy to read but needed some improvement in the discussion section.

Please see my comments below.

Table S1, please add this table as main table 1.

Figure 1: Please indicate in the Y-axis “fold change of miRNA 21”. How the fold change of miRNA 21 expression was calculated is not completely clear. Would you please clarify this information in the figure legend?

Figure 2: Please indicate reference description in Figure 2 for blue and red histograms.

In the result section, please correct the title of the figure, as “Expression of miR-21 targets gene in splenic leukocytes of CanL and healthy dogs ”. Same comment for figure legend 2

Figure 3: Please indicate in Figure legend 3 title, transfection of what is referring to.

In the result section, indicate the reason for using miRNA mimics and inhibitors. Have you been waiting for opposite results?

Figure 4: Please also include data of CD4 and CD8 in Figure 4.

Discussion section.

Even though your results showed no changes in the regulation of FAS, FASL, and CCR7 by miRNA 21, these results could be related to low transfection efficacy, not adequate concentration range of mimic or inhibitor, among others. Did you try different concentrations? Please discuss these aspects very carefully and include them as limitations of the work

This sentence is unclear; please revise “FAS and FASL play critical roles in the immune system, particularly in the death of target cells infected by pathogens and autoreactive lymphocytes' death.

Same comment as above regarding the effect of miRNA 21 mimic and inhibitor on CD4 and CD8. Did you perform a doses resp

Reviewer #2: Dr. Bragato and colleagues have submitted a manuscript entitled: "miRNA-21 regulates CD69 and IL-10 expression in canine leishmaniasis." The study explored the role of miR-21 in dogs with CanL comparing the expression of miR-21 targets in samples from infected and non-infected dogs. Additionally, they performed an in vitro experiment with cells transfected with miR-21 mimics and inhibitors. The authors found higher expression of miR-21 in infected dogs and a decrease expression in miR-21 targets in this population. In vitro inhibition of miR-21 decreased CD69 expression in B lymphocytes and IL-10 in supernatants. The authors concluded miR-21 may be a target for CanL.

Line 101: The authors stated that “Splenic fragments in control dogs were removed by surgical excision [25]”. I wish to understand if the Animal Ethics Committee approved the collection of splenic fragments from healthy animals. If yes, could you please clarify in which circumstances? Please, provide the Animal Ethics certificate. Also, it is unclear why reference 25 was used.

Line 105: Please, explain how the fragments were macerated.

Line 110: Please provide the description or the commercial reference for the red blood cell lysis buffer

Line 136: Confirm if the DNA was extracted from cells or directly from tissue

Line 157: The authors stated “To confirm that miR-21 is upregulated in dogs with CanL obtained by [11],” Please, explain why reference 11 was used.

Line 169: A pool of which cDNAs was used for normalization - control, infected, or both? Please explain where the absolute quantification of miR-21 was used.

Line 173: Please, insert the reference or complete description for SNORD96A

Results

Figure 1 legend: Please, insert the number of animals (samples) analyzed.

Figure 2 legend: Please, detail which range is referring, min-max, SD? I suggest presenting the control group first in the graph (same order as in figure 1).

Figures 4 and 5 legends: Please, detail which range is referring, min-max, SD?

I am concerned about the validation of the transfection. The authors did not show a control condition for the experiment (a condition with the transfection reagent only, without the scrambled sequence). Did you perform a pilot experiment with i.e., cells treated with transfection reagent only? Also, it is mandatory to validate the expression of miR-21 after transfection. How do the authors know the transfection efficiency relative to miR-21 inhibition? Is it possible to run an experiment to evaluate the miRNA expression after transfection? Do the authors have frozen samples? Gene expression should be similar in both untransfected cells and cells transfected with the negative control. Comment, please.

Besides that, the average CD69 expression in the control group was around 40000 (figure 2). When the cells of this group were transfected with the scrambled sequence, the average expression was around 75000 (figure 3). It seems that the scrambled condition is increasing the expression. At the same time, the inhibitor condition is achieving an expression around 40000, similar to the control group expression in figure 2. Results from the negative control should be compared to results from untransfected cells. It seems that adequate controls are missing. Comment, please.

The authors found significantly less expression of CD69 in control and canL groups when miR-21 was mimic on the other side the inhibition did not produce the opposite effect. Comment, please.

Discussion:

Line 334-335: Maybe the low transfection rate of this study does not support the previous sentence “suggesting that, in dogs, these proteins may not be targets of miR-21.” Please, comment.

6. PLOS authors have the option to publish the peer review history of their article (what does this mean?). If published, this will include your full peer review and any attached files.

Reviewer #1: No

Reviewer #2: No

---

## [Author Response · Author response to Decision Letter 0]

30 Sep 2021

Response to Reviewers

Manuscript No.: PONE-D-21-21939

Title: miRNA-21 regulates CD69 and IL-10 expression in canine leishmaniasis

Dear Editor,

We appreciate the careful review of our manuscript by the reviewers. Their constructive comments and suggestions are much appreciated, and the new version of the manuscript has benefited considerably from their assistance Please note that all changes made in the text are highlighted in red.

Editor Comment:

The authors should provide experiments and controls for the transfection efficacy of the micro RNA to be sure that the effect was due to the action or not of the transfected micro RNA;

R: The experiment that represents transfection efficacy was added to supporting information;.

Reviewer Comments:

Author’s answer to Reviewer Comments: We thank the referee for providing constructive comments to improve the article. All criticisms and detailed corrections have been addressed and the following revisions have been made.

We have accepted all the suggestions and requests made by the Editor and Reviewers in the manuscript. 

Sincerely yours,

Valéria Marçal Felix de Lima, MSc, PhD

São Paulo State University – UNESP

Reviewer #1

Author’s answer: We thank the referee for providing constructive comments to improve the article. All criticisms and detailed corrections have been addressed and the following revisions have been made.

Reviewer #1’s comments Answers

Table S1, please add this table as main table 1 

R: Table 1 was put as main table as recommended.

Figure 1: Please indicate in the Y-axis “fold change of miRNA 21”. How the fold change of miRNA 21 expression was calculated is not completely clear. Would you please clarify this information in the figure legend? 

R: The required information was added to the figure legend.

Figure 2: Please indicate reference description in Figure 2 for blue and red histograms. In the result section, please correct the title of the figure, as “Expression of miR-21 targets gene in splenic leukocytes of CanL and healthy dogs”. Same comment for figure legend 2 

R: The information was modified in the text. 

Figure 3: Please indicate in Figure legend 3 title, transfection of what is referring to. In the result section, indicate the reason for using miRNA mimics and inhibitors. Have you been waiting for opposite results? 

R: The legend of the figure was modified. We used both mimics and inhibitor because increased and decreased expression could facilitate the understanding of how miR-21 does the regulation. 

Figure 4: Please also include data of CD4 and CD8 in Figure 4. 

R: The figure was added.

Discussion section. Even though your results showed no changes in the regulation of FAS, FASL, and CCR7 by miRNA 21, these results could be related to low transfection efficacy, not adequate concentration range of mimic or inhibitor, among others. Did you try different concentrations? Please discuss these aspects very carefully and include them as limitations of the work. 

R: We tested all the different concentrations recommended by the manufacturer in order to optimize the transfection rate, and used the concentrations that gave the best results. 

This sentence is unclear; please revise “FAS and FASL play critical roles in the immune system, particularly in the death of target cells infected by pathogens and autoreactive lymphocytes' death. 

R: This sentence was reformulated. 

Same comment as above regarding the effect of miRNA 21 mimic and inhibitor on CD4 and CD8. Did you perform a doses resp

R: Yes, we tested different doses to optimize the results.

Reviewer #2

Author’s answer: We thank the referee for providing constructive comments to improve the article. All criticisms and detailed corrections have been addressed and the following revisions have been made.

Reviewer #2’s comments Answers

Line 101: The authors stated that “Splenic fragments in control dogs were removed by surgical excision [25]”. I wish to understand if the Animal Ethics Committee approved the collection of splenic fragments from healthy animals. If yes, could you please clarify in which circumstances? Please, provide the Animal Ethics certificate. Also, it is unclear why reference 25 was used. 

R: Animal Ethics Committee approved the collection of splenic fragments from healthy animals. The owners assigned a term of consentient authorizing the procedure. Reference 25 was used to describe the surgical procedure. 

Line 105: Please, explain how the fragments were macerated. 

R: The information was added to the text.

Line 110: Please provide the description or the commercial reference for the red blood cell lysis buffer 

R: The information was added to the text.

Line 136: Confirm if the DNA was extracted from cells or directly from tissue

R: DNA was extracted from cells.

Line 157: The authors stated “To confirm that miR-21 is upregulated in dogs with CanL obtained by [11],” Please, explain why reference 11 was used.

R: Reference 11 was used to show that the waiting result has already been observed in other paper. 

Line 169: A pool of which cDNAs was used for normalization - control, infected, or both? Please explain where the absolute quantification of miR-21 was used.

R: A pool of both cDNAs. Absolute quantification was used in all samples of both groups.

Line 173: Please, insert the reference or complete description for SNORD96A.

R: The information was added to the text.

Results

Figure 1 legend: Please, insert the number of animals (samples) analyzed.

Figure 2 legend: Please, detail which range is referring, min-max, SD? I suggest presenting the control group first in the graph (same order as in figure 1).

Figures 4 and 5 legends: Please, detail which range is referring, min-max, SD?

R: All modifications were done.

I am concerned about the validation of the transfection. The authors did not show a control condition for the experiment (a condition with the transfection reagent only, without the scrambled sequence). Did you perform a pilot experiment with i.e., cells treated with transfection reagent only? Also, it is mandatory to validate the expression of miR-21 after transfection. How do the authors know the transfection efficiency relative to miR-21 inhibition? Is it possible to run an experiment to evaluate the miRNA expression after transfection? Do the authors have frozen samples? Gene expression should be similar in both untransfected cells and cells transfected with the negative control. Comment, please.

According the manufacturer, “The expression of an endogenous gene, which is known to be a target of the miRNA under study, is measured after mimic/inhibitor transfection. The effect of the mimic/inhibitor is determined by comparing this result with the gene expression in untransfected cells or cells transfected with a negative control. Gene expression is often measured at the protein level, for example, by western blot, as miRNAs often inhibit translation of their target genes and do not cause degradation of the target transcript. This means that the effect of a miRNA mimic or inhibitor can often not be determined using quantitative, real-time PCR.”

Besides that, the average CD69 expression in the control group was around 40000 (figure 2). When the cells of this group were transfected with the scrambled sequence, the average expression was around 75000 (figure 3). It seems that the scrambled condition is increasing the expression. At the same time, the inhibitor condition is achieving an expression around 40000, similar to the control group expression in figure 2. Results from the negative control should be compared to results from untransfected cells. It seems that adequate controls are missing. Comment, please.

Although the numbers are different, the analysis was performed and there was no statistical difference. 

According to manufacturer, “A negative control should be transfected in every experiment and will indicate if results are nonspecific. Comparison of results from the negative control with results from the miRNA mimic under study can be used to confirm that the observed results are specific to the miRNA mimic under study.”

The graph below represents the control group containing comparative analyzes with non-transfected cells and using only the transfection reagent (HiperFect), demonstrating that there was no statistical difference between the other groups. 

The authors found significantly less expression of CD69 in control and canL groups when miR-21 was mimic on the other side the inhibition did not produce the opposite effect. Comment, please.

The amount of miR-21 present in the samples can be so high that the concentration used in the inhibitor and indicated by the manufacturer was not enough to show an opposite effect. 

Discussion:

Line 334-335: Maybe the low transfection rate of this study does not support the previous sentence “suggesting that, in dogs, these proteins may not be targets of miR-21.” Please, comment.

The low transfection rate obtained may not have been sufficient to modulate the expression of these molecules.

In Other experimental models, these proteins are targets of miR-21:

1- Liu Y, Ren L, Liu W, Xiao Z. MiR-21 regulates the apoptosis of keloid fibroblasts by caspase-8 and the mitochondria-mediated apoptotic signaling pathway via targeting FasL. Biochem Cell Biol. 2018 Oct;96(5):548-555. doi: 10.1139/bcb-2017-0306. Epub 2018 Mar 10. PMID: 29527928.

2- Shang C, Guo Y, Hong Y, Liu YH, Xue YX. MiR-21 up-regulation mediates glioblastoma cancer stem cells apoptosis and proliferation by targeting FASLG. Mol Biol Rep. 2015 Mar;42(3):721-7. doi: 10.1007/s11033-014-3820-3. Epub 2014 Nov 14. PMID: 25394756.

3- Marega LF, Teocchi MA, Dos Santos Vilela MM. Differential regulation of miR-146a/FAS and miR-21/FASLG axes in autoimmune lymphoproliferative syndrome due to FAS mutation (ALPS-FAS). Clin Exp Immunol. 2016 Aug;185(2):148-53. doi: 10.1111/cei.12800. Epub 2016 May 24. PMID: 27060458; PMCID: PMC4954998.

4- Sayed D, He M, Hong C, Gao S, Rane S, Yang Z, Abdellatif M. MicroRNA-21 is a downstream effector of AKT that mediates its antiapoptotic effects via suppression of Fas ligand. J Biol Chem. 2010 Jun 25;285(26):20281-90. doi: 10.1074/jbc.M110.109207. Epub 2010 Apr 19. PMID: 20404348; PMCID: PMC2888441.

---

## [Decision Letter · Decision Letter 1]

26 Oct 2021

PONE-D-21-21939R1miRNA-21 regulates CD69 and IL-10 expression in canine leishmaniasisPLOS ONE

Dear Dr. de Lima,

Thank you for submitting your manuscript to PLOS ONE. After careful consideration, we feel that it has merit but does not fully meet PLOS ONE’s publication criteria as it currently stands. Therefore, we invite you to submit a revised version of the manuscript that addresses the points raised during the review process: 1) Review the statistic accordingly to reviewer #2 comments;2) Please, answer to the comments raised by both reviewers.

We look forward to receiving your revised manuscript.

Kind regards,

Paulo Lee Ho, Ph.D.

Academic Editor

PLOS ONE

Reviewers' comments:

Reviewer's Responses to Questions

**Comments to the Author**

1. If the authors have adequately addressed your comments raised in a previous round of review and you feel that this manuscript is now acceptable for publication, you may indicate that here to bypass the “Comments to the Author” section, enter your conflict of interest statement in the “Confidential to Editor” section, and submit your "Accept" recommendation.

Reviewer #1: (No Response)

Reviewer #2: (No Response)

2. Is the manuscript technically sound, and do the data support the conclusions?

Reviewer #1: Yes

Reviewer #2: Partly

3. Has the statistical analysis been performed appropriately and rigorously? 

Reviewer #1: Yes

Reviewer #2: No

4. Have the authors made all data underlying the findings in their manuscript fully available?

Reviewer #1: Yes

Reviewer #2: No

5. Is the manuscript presented in an intelligible fashion and written in standard English?

Reviewer #1: Yes

Reviewer #2: Yes

6. Review Comments to the Author

Reviewer #1: The authors have made most of the solicited changes, but some additional changes are still needed for a clear presentation of the data. This reviewer considers that the paper can be accepted after performing the following suggested corrections.

Figure 1: Include “fold change of miRNA 21” on the Y-axis of the Graph. This will help the reader to quickly understand what fold changes are referring to in that graph.

Figure 4: The Y-axis of the 3 Graphs showing data of CD69 expression on CD4, CD8, and CD21 positive cells are incorrectly labeled. Those graphs should state “Mean Fluorescence Intensity of CD69 on CD4+ cells, or CD8+ cells or CD21+ cells” instead of the current labels that are very confusing, Mean CD8+ cells (FL1)/CD69+ cells (FL2). Please correct accordingly.

S2 Fig. It is not clear what the authors are measuring as the control of transfection. What do more dead cells (7ADD+) mean in regard to transfection efficacy? please clarify this point in result section.

Reviewer #2: The authors answered all points addressed by the reviewer. Unfortunately, the authors were not able to solve the main issue raised by the reviewer. Although the author affirms that there is no difference among scrambled, untransfected, and HiperFect conditions, only in two samples in the scrambled condition the CD69 expression is high. So, my point is that the mimic is not decreasing the expression, but something happens in the scrambled that resulted in a higher expression. I still wish to see the raw data of all of them.

I strongly suggest a statistical review as in my checking (with the raw data available) there were no defferences in CD69 expression in control and CanL groups (ANOVA followed by Dunn's multiple comparison test).

7. PLOS authors have the option to publish the peer review history of their article (what does this mean?). If published, this will include your full peer review and any attached files.

Reviewer #1: No

Reviewer #2: No

---

## [Author Response · Author response to Decision Letter 1]

29 Oct 2021

The response to reviewers is also attached as a Microsoft Word doc.

Response to Reviewers

Manuscript No.: PONE-D-21-21939

Title: miRNA-21 regulates CD69 and IL-10 expression in canine leishmaniasis

Dear Editor,

We appreciate the careful review of our manuscript by the reviewers. Their constructive comments and suggestions are much appreciated, and the new version of the manuscript has benefited considerably from their assistance Please note that all changes made in the text are highlighted in red.

Reviewer Comments:

Author’s answer to Reviewer Comments: We thank the referee for providing constructive comments to improve the article. All criticisms and detailed corrections have been addressed and the following revisions have been made.

We have accepted all the suggestions and requests made by the Reviewers in the manuscript. 

Sincerely yours,

Valéria Marçal Felix de Lima, MSc, PhD

São Paulo State University – UNESP

Reviewer #1

Author’s answer: We thank the referee for providing constructive comments to improve the article. All criticisms and detailed corrections have been addressed and the following revisions have been made.

Reviewer #1’s comments Answers

Figure 1: Include “fold change of miRNA 21” on the Y-axis of the Graph. This will help the reader to quickly understand what fold changes are referring to in that graph. 

R: The y-axis was renamed as recommended.

Figure 4: The Y-axis of the 3 Graphs showing data of CD69 expression on CD4, CD8, and CD21 positive cells are incorrectly labeled. Those graphs should state “Mean Fluorescence Intensity of CD69 on CD4+ cells, or CD8+ cells or CD21+ cells” instead of the current labels that are very confusing, Mean CD8+ cells (FL1)/CD69+ cells (FL2). Please correct accordingly. 

R: The Y-axis title was modified. 

S2 Fig. It is not clear what the authors are measuring as the control of transfection. What do more dead cells (7ADD+) mean in regard to transfection efficacy? please clarify this point in result section.

R: The information to better explain the transfection rates was added in the methods section. 

Reviewer #2

Author’s answer: We thank the referee for providing constructive comments to improve the article. All criticisms and detailed corrections have been addressed and the following revisions have been made.

Reviewer #2’s comments Answers

The authors answered all points addressed by the reviewer. Unfortunately, the authors were not able to solve the main issue raised by the reviewer. Although the author affirms that there is no difference among scrambled, untransfected, and HiperFect conditions, only in two samples in the scrambled condition the CD69 expression is high. So, my point is that the mimic is not decreasing the expression, but something happens in the scrambled that resulted in a higher expression. I still wish to see the raw data of all of them. I strongly suggest a statistical review as in my checking (with the raw data available) there were no defferences in CD69 expression in control and CanL groups (ANOVA followed by Dunn's multiple comparison test).

CD69 expression in untransfected cells in CanL and Control group

Normality test (don’t pass)

Non parametric test to compare the expression in CanL and Control group (Mann Whitney test).

Analysis post transfection with mimics and inhibitors of miR-21:

CanL group data

Normality test (don’t pass)

Friedman test (non parametric)

Post test (Dunn’s multiple comparisons)

Control Group data

Normality test

Friedman test

Dunn’s multiple comparison test

---

## [Decision Letter · Decision Letter 2]

24 Nov 2021

PONE-D-21-21939R2miRNA-21 regulates CD69 and IL-10 expression in canine leishmaniasisPLOS ONE

Dear Dr. de Lima,

Thank you for submitting your manuscript to PLOS ONE. After careful consideration, we feel that it has merit but does not fully meet PLOS ONE’s publication criteria as it currently stands. Therefore, we invite you to submit a revised version of the manuscript that addresses the points raised during the review process.

1) Provide the statistical analysis as requested by the reviewer #2.2) Please, answer all the comments raised by both the reviewers.

We look forward to receiving your revised manuscript.

Kind regards,

Paulo Lee Ho, Ph.D.

Academic Editor

PLOS ONE

Journal Requirements:

Reviewers' comments:

Reviewer's Responses to Questions

**Comments to the Author**

1. If the authors have adequately addressed your comments raised in a previous round of review and you feel that this manuscript is now acceptable for publication, you may indicate that here to bypass the “Comments to the Author” section, enter your conflict of interest statement in the “Confidential to Editor” section, and submit your "Accept" recommendation.

Reviewer #1: All comments have been addressed

Reviewer #2: (No Response)

2. Is the manuscript technically sound, and do the data support the conclusions?

Reviewer #1: Yes

Reviewer #2: Partly

3. Has the statistical analysis been performed appropriately and rigorously? 

Reviewer #1: Yes

Reviewer #2: No

4. Have the authors made all data underlying the findings in their manuscript fully available?

Reviewer #1: Yes

Reviewer #2: Yes

5. Is the manuscript presented in an intelligible fashion and written in standard English?

Reviewer #1: Yes

Reviewer #2: Yes

6. Review Comments to the Author

Reviewer #1: The authors have included all suggestions and clarified pending questions. I do not have further comments.

Reviewer #2: I strongly suggest (again) a revision in the statistics of the results in figure 3.

Data of different conditions (mimic, inhibitor, scrambled, hiperfect, and untransfected) must be analyzed by ANOVA followed by multicomparison's posttest for canine and control groups.

I guess you used Dunn's multiple comparisons test to compare the mean rank of each condition with the mean rank of a control group. Did you use scrambled as a control? Why didn’t you compare the mean rank of each column with the mean rank of every other column?

I checked the statistics with the data provided using Prism version 8.2.1.

Please, send me the Prism file.

7. PLOS authors have the option to publish the peer review history of their article (what does this mean?). If published, this will include your full peer review and any attached files.

Reviewer #1: No

Reviewer #2: No

---

## [Author Response · Author response to Decision Letter 2]

30 Nov 2021

We appreciate the reviewers' suggestions. All of them have been answered. The file with the requested statistic data was sent by email.

---

## [Decision Letter · Decision Letter 3]

4 Jan 2022

PONE-D-21-21939R3miRNA-21 regulates CD69 and IL-10 expression in canine leishmaniasisPLOS ONE

Dear Dr. de Lima,

Thank you for submitting your manuscript to PLOS ONE. After careful consideration, we feel that it has merit but does not fully meet PLOS ONE’s publication criteria as it currently stands. Therefore, we invite you to submit a revised version of the manuscript that addresses the points raised during the review process.

We look forward to receiving your revised manuscript.

Kind regards,

Paulo Lee Ho, Ph.D.

Academic Editor

PLOS ONE

Reviewers' comments:

Reviewer's Responses to Questions

**Comments to the Author**

1. If the authors have adequately addressed your comments raised in a previous round of review and you feel that this manuscript is now acceptable for publication, you may indicate that here to bypass the “Comments to the Author” section, enter your conflict of interest statement in the “Confidential to Editor” section, and submit your "Accept" recommendation.

Reviewer #2: (No Response)

2. Is the manuscript technically sound, and do the data support the conclusions?

Reviewer #2: Partly

3. Has the statistical analysis been performed appropriately and rigorously? 

Reviewer #2: No

4. Have the authors made all data underlying the findings in their manuscript fully available?

Reviewer #2: No

5. Is the manuscript presented in an intelligible fashion and written in standard English?

Reviewer #2: Yes

6. Review Comments to the Author

Reviewer #2: According to the Prism file provided by the authors, the statistical analysis of CD69 expression after transfection must be performed with a parametric test (ANOVA) for the CONTROL group instead of the nonparametric test used. Also, data of HiperFect buffer and untransfected cells must be provided in figure 3. Update methods and results section accordingly.

7. PLOS authors have the option to publish the peer review history of their article (what does this mean?). If published, this will include your full peer review and any attached files.

Reviewer #2: No

---

## [Author Response · Author response to Decision Letter 3]

13 Feb 2022

Figure 3 was modified and HiperFect and untransfected cells data were added.

To claim that our data has a normal distribution, we rigorously pass three different normality tests. Since of the three, only 1 test showed normality in the data, while the other 2 showed that n is too small to present a normal distribution. Given this, we chose to use a non-parametric test, which best represents our data.

Similar experiment has already been published by our research group, using this same statistical test, in the work of MELO, et al., 2019.

Attached are some links to works with values of n similar to ours, which used non-parametric tests.

https://www.ncbi.nlm.nih.gov/pmc/articles/PMC4029133/

https://www.ncbi.nlm.nih.gov/pmc/articles/PMC7023135/

https://www.ncbi.nlm.nih.gov/pmc/articles/PMC6905561/

https://www.ncbi.nlm.nih.gov/pmc/articles/PMC5962361/

https://www.ncbi.nlm.nih.gov/pmc/articles/PMC6962187/

---

## [Decision Letter · Decision Letter 4]

28 Feb 2022

miRNA-21 regulates CD69 and IL-10 expression in canine leishmaniasis

PONE-D-21-21939R4

Dear Dr. de Lima,

We’re pleased to inform you that your manuscript has been judged scientifically suitable for publication and will be formally accepted for publication once it meets all outstanding technical requirements.

Kind regards,

Paulo Lee Ho, Ph.D.

Academic Editor

PLOS ONE

Additional Editor Comments (optional):

Reviewers' comments:

Reviewer's Responses to Questions

**Comments to the Author**

1. If the authors have adequately addressed your comments raised in a previous round of review and you feel that this manuscript is now acceptable for publication, you may indicate that here to bypass the “Comments to the Author” section, enter your conflict of interest statement in the “Confidential to Editor” section, and submit your "Accept" recommendation.

Reviewer #2: All comments have been addressed

2. Is the manuscript technically sound, and do the data support the conclusions?

Reviewer #2: Yes

3. Has the statistical analysis been performed appropriately and rigorously? 

Reviewer #2: N/A

4. Have the authors made all data underlying the findings in their manuscript fully available?

Reviewer #2: No

5. Is the manuscript presented in an intelligible fashion and written in standard English?

Reviewer #2: Yes

6. Review Comments to the Author

Reviewer #2: (No Response)

7. PLOS authors have the option to publish the peer review history of their article (what does this mean?). If published, this will include your full peer review and any attached files.

Reviewer #2: No

---

## [Editor Report · Acceptance letter]

9 Mar 2022

PONE-D-21-21939R4 

miRNA-21 regulates CD69 and IL-10 expression in canine leishmaniasis 

Dear Dr. de Lima:

I'm pleased to inform you that your manuscript has been deemed suitable for publication in PLOS ONE. Congratulations! Your manuscript is now with our production department. 

Kind regards, 

on behalf of

Dr. Paulo Lee Ho 

Academic Editor

PLOS ONE